

# Direct local building inundation depth determination in 3D point clouds generated from user-generated flood images

Luisa Griesbaum[1], Sabrina Marx[1], Bernhard Höfle[1,2]

[1]GIScience, Department of Geography, Heidelberg University, Heidelberg, 69120, Germany
[2]Heidelberg Center for the Environment (HCE), Heidelberg University, 69120 Heidelberg, Germany

*Correspondence to*: Luisa Griesbaum (griesbaum@uni-heidelberg.de)

**Abstract.** In recent years, the number of people affected by flooding caused by extreme weather events has increased considerably. In order to provide support in disaster recovery or to develop mitigation plans, accurate flood information is necessary. Particularly pluvial urban floods, characterized by high temporal and spatial variations, are not well documented.

This study proposes a new, low-cost approach to determining local flood elevation and inundation depth of buildings based on user-generated flood images. It first applies close-range digital photogrammetry to generate a geo-referenced 3D point cloud. Second, based on estimated camera orientation parameters, the flood level captured in a single flood image is mapped to the previously derived point cloud. The local flood elevation and the building inundation depth can then be derived automatically from the point cloud. The proposed method is carried out once for each of 66 different flood images showing the same building

façade. An overall accuracy of 0.05 m with an uncertainty of ± 0.13 m for the derived flood elevation within the area of interest and an accuracy of 0.13 m ± 0.10 m for the determined building inundation depth is achieved. Our results demonstrate that the proposed method can provide reliable flood information on a local scale using user-generated flood images as input. The approach can thus allow inundation depth maps to be derived even in complex urban environments with relatively high accuracies.

**1 Introduction**

World-wide the number of extreme weather events has increased in recent years (CRED - Centre for Research on the Epidemiology of Disasters, 2016). The reasons for this accumulation of flood events are numerous: On the one hand, climate change might be responsible for variations in weather events. On the other, land-use changes such as increased ground surface sealing are leading to uncontrolled overland runoff and rainwater drainage, especially in urban areas (Douglas et al., 2010;

Mason et al., 2014). Due to spreading urbanization, more and more of the areas at high risk of flooding have become populated, for example regions close to rivers or at the foot of hills. Since this leads to larger numbers of people being affected in terms of physical or monetary damages, or even human costs, there is a major need for urban flood-risk management (Zevenbergen et al., 2008; Hammond et al., 2013; Iervolino et al., 2015). Information about previous floods, such as flood elevation and local



inundation depths, are of high relevance for mitigation and resilience planning to assess and minimize the impact of disastrous events.

Urban flood events can be differentiated according to their major causes into the following groups: fluvial flooding (e.g., flash floods, river-based urban floods), groundwater flooding, coastal flooding, and pluvial urban flooding. Thus far, the main

traditional data sources for monitoring and documenting floods are gauge-system measurements, forecasted and measured precipitation rates, and information derived from remote sensing techniques, such as satellite imagery or Light Detection and Ranging (LiDAR) (Lo et al., 2015).

Pluvial floods are often triggered by blocked or overburdened sewage systems in combination with heavy rainfalls (Maksimović et al., 2009; Hammond et al., 2013). They are highly dynamic phenomena with high spatial and temporal

variation (Blanc et al., 2012). Most of the above-mentioned traditional techniques are, thus, not suitable because of their relatively coarse spatial and temporal resolution. Gauge systems usually do not cover a city's whole street network, and precipitation rates are generally not sufficient for the simulation of local pluvial floods. Furthermore, detailed remote-sensing data are typically not available at short notice. Thus, many studies utilize remote sensing data in the aftermath of an event for post event flood simulation in order to retrieve the deluge extent or flood-water depth of previous events (Bates et al., 2003;

Schumann et al., 2008; Abdullah et al., 2009; Chen et al., 2009; Schumann et al., 2011; Merkuryeva et al., 2015). Only a few studies focus on flood-level and depth determination from flood data acquired during the event itself (Matgen et al., 2007; Mason et al., 2010; Mason et al., 2014; Iervolino et al., 2015). Matgen et al. (2007) report an RMSE of 0.41 m for flood elevation along a 1 km river section by combining high-precision digital elevation models (DEM) with flood extent maps from SAR data. Iervolino et al. (2015) derive local building inundation from SAR data with accuracies of 0.24–0.81 m. However,

the Flood Loss Estimation Model for the Private Sector (FLEMOps) requires building inundation depth accuracies around 0.10 m for flood damage assessment, since it suggests damage classification according to the inundation depth of a building in steps of 0.20 m to 0.50 m (Thieken, 2008).

Thus, in order to improve flood disaster management in response to urban flooding, more detailed information on pluvial urban floods in terms of spatial and temporal resolution is necessary (Hammond et al., 2013:14; Merkuryeva et al., 2015:79). The

combination of the high temporal dynamics of the phenomenon and the need for high-resolution, on-demand, in situ data make it particularly difficult to measure urban flooding. It is therefore necessary to develop new methods to generate water-level information about pluvial urban floods using available high-resolution data (Price and Vojinovic, 2008:261). One attempt to achieve higher resolution and easier availability of pluvial flood data in urban areas is to apply close-range photogrammetry (CRP), i.e. a sequence of digital image processing methods based on computer vision algorithms (e.g., Structure from Motion;

SfM), and photogrammetric approaches (e.g., Dense Matching; DM) to derive 3D point clouds or high-resolution Digital Terrain Models (DTM) (Meesuk et al., 2015; Shaad et al., 2016). Smith et al. (2014) demonstrate the potential of using photogrammetric point clouds for the reconstruction of high water marks of a flash flood event at a river channel. However, in urban areas, such high water marks (i.e. clearly visible flood relics like mud lines) are typically removed very quickly after the flood event. Thus, typically only very few single images document the actual flood elevation in urban settings.



To complement traditional documentation systems and to tackle their temporal, spatial, or cost related limitations, a possible approach can be the use of User Generated Content (UGC), such as Ambient Geographic Information (AGI) (Stefanidis et al., 2013) or Volunteered Geographic Information (VGI) (Goodchild, 2007). The increasing distribution of mobile devices, in conjunction with the ever-expanding use of Web 2.0, has led to more virtual participation in flood mitigation activities as well as in flood event documentation (Fazeli et al., 2015; Klonner et al., 2016). Subsequently, many urban flood events are now indirectly documented by means of user-generated, partially geotagged flood images, posted on social media platforms. Various studies investigate the feasibility and benefits of using these new data sources for flood management (Fazeli et al., 2015). McDougall and Temple-Watts (2012), and Fohringer et al. (2015) successfully demonstrate the potential of VGI data for flood reconstruction by manual in-field measurements of the flood elevation, given in flood images. Other studies thereto propose semi-automatic approaches to derive the flood extent or level shown in flood images: Triglav-Čekada and Radovan (2013) map the extent of flooded areas based on geo-located flood images by applying a method where the absolute orientation of an image is found by fitting that image to the superimposed 3D points of a DTM. Narayana et al. (2014) propose a technique to determine building inundation depth by matching a manually traced flood line from a given flood image to a respective non-flood image with the help of corresponding image features. However, this methodology has not been tested in a real-world set-up and requires a priori knowledge about the buildings' height in order to determine the inundation depth. Furthermore, the approach uses information derived from 2D imagery, which has inherent restrictions in terms of perspective.

From these studies it emerges that there is still a lack of automatic approaches that allow singular flood-event-based information to be extracted from unstructured user-generated images in order to reconstruct flood parameters at a local building scale in 3D. The aim of our study is to develop a low-cost method to extract local flood elevation as well as building inundation depth in urban settings on the basis of ordinary user-generated photographs. This semi-automatic workflow includes automatic flood level detection from flood images, as well as a new and innovative way to integrate singular, i.e. flood-event-based, information provided by a single flood image into a photogrammetric point cloud by extending existing methods in order to reconstruct the flood elevation. In contrast to previous photogrammetric approaches where two or more perspectives are necessary to reconstruct a given object in 3D, our method is a two stage approach, where 1) the 3D scene is reconstructed independent of flood images (before or after the flood event), and 2) the single flood image information is integrated into the 3D scene in order to reconstruct the flood level in 3D.

## 2 Study area

The area chosen for study is located at the Karl-Theodor Bridge in Heidelberg, Germany, on the river Neckar at 49.41334° N and 8.70996° E. The Neckar flows northwards between the Swabian Jura and the Black Forest into the Rhine River at Mannheim, and it drains major parts of the German Federal State Baden-Württemberg. The closest gauge station is located about 4 km upstream of the study area. The chosen area (Fig. 1) is characterized by a declined road section parallel to the river that leads below the bridge. It is continually at risk of river-based urban floods and regularly inundated.



The flood event examined in this study occurred on May 30th, 2016. Several days of heavy rainfall led to gauge measurements reaching almost 430 cm (200 cm is the normal gauge reading). According to the discharge curve, the peak water elevation was noted between 16:00 h and 23:00 h, after which the gauge reading started to decline (LUBW - Landesanstalt für Umwelt, Messungen und Naturschutz Baden-Württemberg, 2016). The area experienced an overflow of the riverbed, which caused the

inundation of the nearby roads Neckarstaden and Am Hackteufel (Fig. 1). The flooding reached the facing side of the adjacent houses, which comprises the central object of interest in this research.

## 3 Data sets

### 3.1 Flood images

Of primary importance for this study are flood images showing the inundated object of interest, photographed during the flood

event on the 30th of May 2016. Image acquisition took place using two mobile devices, as are typically used for imagery contributed to social media, from different randomly chosen and accessible positions around the object of interest: 1) at around 16:00 h and 19:30 h local time with a Samsung Galaxy A3 mobile phone camera with image resolutions of 3264 x 2448 pixels and 3264 x 1836 pixels and 2) at around 18:30 h local time on the same day with a Samsung Galaxy S2 mobile phone camera with a resolution of 2560 x 1920 pixels. In total, 66 flood images showing the object of interest were captured using

automatically set camera parameters.

### 3.2 Non-flood images

Terrestrial non-flood images showing the object of interest were captured in June 2016, after the flood event had subsided, with a Sony Alpha 57 16-megapixel single-lens translucent (SLT) camera, which provides images with a resolution of 4912 x 3264 pixels. The camera settings were automatically determined by the device. Image acquisition took place with the

camera's perspective converging towards the study area and an approximate distance of 3.5 m between the individual camera positions (Fig. 1). At almost all of the 25 camera positions, two or more images were required to capture the whole building's façade. In total, 63 non-flood images were captured.

### 3.3 Reference data

A high-end state of the art Terrestrial Laser Scanning (TLS) measurement system Riegl VZ-400 is used to provide reference

data for the analyses. The TLS data were captured in May 2016, before the severe flooding had occurred. The scanning system operates with a wavelength of 1550 nm and a beam divergence of 0.35 mrad. Range precision (repeatability) and accuracy (conformity of measurements to actual geometry) are 3 mm and 5 mm at 100 m, respectively, as given by the manufacturer's datasheet (Riegl, 2016). The scene was captured from four different scanning positions. When combined into a single dataset, i.e. when co-registered, the overlapping scans result in a point cloud with a total of 29 million measurements within the area

of interest. The registration accuracy is determined via 64 point pairs manually picked in the individual scans. Point pair



distances for $x$, $y$, and $z$ are between 0.001 m ($x$), 0.001 m ($y$), and 0.000 m ($z$), with a standard deviation between 0.020 m ($x$), 0.010 m ($y$), and 0.010 m ($z$).

The reference absolute flood elevation at the given time of the flood event is determined on the basis of a sequence of images of a near-by staff gauge (ca. 20 m to the object of interest). The median water level elevation derived by averaging this image sequence is $Z_{w,TLS} = 153.83$ m a.s.l. with an amplitude of ± 0.10 m reflecting water undulation and waves.

Additionally, in the aftermath of the flood event, the building inundation depth is also measured in the field at seven distinct positions along the building's façade (Fig. 2). An example flood image captured at 16:00 h local time serves as reference for the manual measurements. Table 1 gives an overview of the captured inundation depth values. Since the mean distance between the minimally and maximally measured inundation depths for all seven positions is 0.10 m, the theoretical uncertainty of the measurements is ± 0.05 m.

Further complementary reference data for the inundation depth are provided by independent expert measurements within the TLS point cloud. Eight experts in 3D point cloud processing from the GIScience Research Group measured inundation depth values at the seven reference positions (Fig. 2).

## 4 Methods

The aim of the proposed method is to derive from a single user-generated flood image a set of 3D points representing 1) the absolute flood elevation ($Z_w$) within the area of interest, and 2) information about the local building inundation depth ($h$). The approach is based on free and open source software solutions and is designed to work based on crowdsourced images of locally scaled urban flood events. It succeeds where other remote sensing techniques fail due to the unsuitability of their spatial and temporal resolutions. The workflow given in Fig. 3 depicts all major steps of the methodology.

### 4.1 Data pre-processing

For each of the 66 flood images, a dense 3D CRP point cloud is derived from a combination of 1) the 63 non-flood input images and 2) the single flood image. A CRP approach generally comprises multiple steps. After detecting and matching similar features in overlapping images, the relative positions of the images as well as the exterior orientation of the cameras used are estimated. At the same time, bundle block adjustment serves to optimize these camera parameters before 3D coordinates of the matched features are derived via ray intersection, resulting in a sparse point cloud. As a final step, dense matching is performed, whereby 3D coordinates of all visible pixels are derived (Eltner et al., 2016). The applied CRP methods provide camera positions and orientations for all of the employed images. In order to provide a highly detailed evaluation of our results, we geo-reference the CRP point cloud based on seven distinct and equally spread out ground control points (GCP) derived from the highly accurate TLS point cloud. Fine registration is then performed with the Iterative Closest Point (ICP) algorithm (Besl and McKay, 1992; Chen and Medioni, 1992) to improve the alignment result. The parameters used to assess the overall quality of the photogrammetric point clouds in comparison to the TLS reference data are explained below.





1) The alignment quality of the photogrammetric point cloud to the TLS reference point cloud is based on the nearest neighboring point between the two point clouds.

2) Completeness and point density are individually determined for the façade plane as well as the terrain plane of the 3D point cloud. The completeness is calculated as a ratio of the number of 0.20 m x 0.20 m cells with a minimum of one point in relation

to the full count of cells within the area of interest (Rosnell and Honkavaara, 2012). The point density is defined as the median of the point count of all 0.20 m x 0.20 m cells (Kraus et al., 2006).

### 4.2 2D waterline detection

The 2D waterline can be described as the demarcation line between water and those parts of image where objects remain above water. In order to trace this demarcation line, the image pixels are categorized into two relevant classes, designated as *water*

and *background*. To this end, two different techniques are proposed: 1) a semi-automatic approach using a supervised machine-learning algorithm for image segmentation and 2) manual image classification. The manual classification results serve as ground truth data for the evaluation of the automated segmentation.

Similar to the work of Bruinink et al. (2015), the semi-automated segmentation approach is based on a trained Random Forest (RF) classifier. As conducted by Marx et al. (2016), 10 % of the available flood images are randomly chosen as training data

before the whole dataset of 66 flood images is segmented by the algorithm. The resulting probability maps are further processed by applying a probability threshold (= 60 %) to assign each pixel to a class, thus generating binary images for the classes of interest (Fig. 4). Residual salt-and-pepper effects as well as small data gaps are removed via a succession of binary opening and closing. The waterbody is then identified by the system as being the largest connected component of pixels classified as water. After semi-automatic as well as manual image segmentation and extraction of water areas, the demarcation line between

water and background, i.e. the 2D waterline, is identified as a sequence of image pixel coordinates. For each pixel column, the image-based coordinates of the upper-most pixel belonging to the water class is assigned as part of the 2D waterline (Bruinink et al., 2015:428).

### 4.3 2D-3D mapping

In order to derive the absolute flood elevation ($Z_w$) within the area of interest as well as the inundation depths along the

25 building's façade ($h$) with full 3D information, the derived 2D waterline image pixel coordinates are mapped to the respective 3D point cloud. This 2D-3D mapping of the 2D waterline pixels is based on photogrammetric principles to reconstruct 3D scenes and thus dependent on the individual flood image's camera position and orientation. Knowing these, the relationship between a 3D point coordinate ($x$, $y$, $z$) in the dense CRP point cloud and the 2D coordinates of its projection onto an image ($u$, $v$) can be formulated as shown in Eq. (1) (Furukawa and Ponce, 2010).

$$d \begin{pmatrix} u \\ v \\ 1 \end{pmatrix} = P \begin{pmatrix} x \\ y \\ z \\ 1 \end{pmatrix} \tag{1}$$





$P$ is a 3x4 projection matrix and $d$ denotes the depth of a point in relation to the image's camera position $C$. $P$ and $C$ are readily available for each image because the CRP approach was applied to prepare the initial 3D point clouds. In order to reconstruct singular image features such as the 2D waterline, which is only given in one flood image, additional (external) geometrical information is applied to derive the depth $d$ and, thus, uniquely reconstruct that feature in 3D. In this study, the additional

5 geometrical information is already known because the targeted 3D water-level points necessarily lie within the building's façade. Consequently, the final 3D water-level points can be located at the intersection point between the calculated line of sight resulting from Eq. (1) and the plane of the façade.

Therefore, the building's façade needs to be identified within the point cloud first. To this end, the 3D point cloud is disjointed into segments representing single planes. Façades can be identified by means of feature constraints such as size (they are

10 usually the biggest, highest, or longest segments), direction (based on the vertical orientation of walls), and topology (the façade plane typically intersects with the terrain plane) (Pu and Vosselman, 2006; Xiao et al., 2008; Serna et al., 2016). In accordance with these criteria for façade identification, in this study, the façade is defined as being the largest connected vertical plane segment within the area of interest.

### 4.4 Flood elevation determination

Since the water's surface can be understood as a continuous rather than a discrete phenomenon, spatially isolated points are eliminated by fitting a linear least squares model with a random sample consensus (RANSAC) algorithm to the preliminarily derived 3D water-level points. The measured water undulation of 0.10 m serves as the threshold. The final set of 3D water-level points is then used for approximation of flood elevation within the area of interest. Due to the extent of the study area (ca. 30 m x 60 m) and the flood's characteristics, the water surface is considered to be perfectly horizontal along the building's

façade in case of a calm water surface. To this end, the statistical distribution of $z$-values of all 3D water-level points is assumed to reflect the actual flood elevation. The quality of the derived flood elevation $Z_w$ is then compared to and evaluated against the reference flood elevation as derived from the nearby staff gauge ($Z_{w,TLS}$ = 153.83 m a.s.l. ± 0.10 m).

### 4.5 Building inundation depth determination

The inundation depth is determined by calculating the distance between the water's surface, i.e. the water-level points derived

in the previous step, and the corresponding terrain elevation at the seven reference positions (Fig. 2). In this study, use of a raster DTM as terrain reference representing the ground surface is demonstrated, however, it can be applied to further terrain models such as points, planes, or meshes.

In order to account for data gaps, as a preparatory step for the DTM generation, terrain points are identified. To this end, a minimum raster at a much coarser scale, namely with a cell size of 5 m x 5 m, exceeding the size of the data gaps, serves as

the initial terrain model. All points within a vertical buffer zone of this coarse DTM are assigned as terrain points. In this case, a threshold of 0.5 m is found to produce the most stable result. The remaining terrain points serve, then, as input for raster generation with a cell size of 0.2 m x 0.2 m and are based on the minimum $z$-value within each cell.





The building inundation depth can be calculated as the vertical off-set between the terrain reference and each of the derived 3D water-level points. In order to compare and evaluate these derived depth measurements against the manual field measurements ($h_{field,R1-R7}$), captured at seven reference positions, only measurements within a horizontal range of ± 20 cm of each reference position are considered.

## 5 Results and discussion

### 5.1 Pre-processing

After geo-referencing of the 66 point clouds, each CRP point cloud is compared to the TLS reference data. This step is done to ensure detailed validation of the final results. It revealed an average cloud-to-cloud median distance of 0.02 m and an average completeness of 37.6 % at the façade and 87.6 % at the terrain (Table 2). These completeness rates are mainly based on data gaps due to occlusion effects caused by parking cars. In summary, the photogrammetric point cloud lacks completeness at a few regions of interest, i.e. terrain and facade, yet shows satisfactory performance concerning geo-referencing which is important only for validation of the results. Therefore, these findings are to be considered when discussing the overall inundation results.

### 5.2 2D waterline detection

The image segmentation resulted in binary images representing the extracted waterbody (Fig. 4). The average classification precision of all images for the water class is 98.5 %, and the average recall is 83.7 %. This means that the detected water pixels are classified with a high degree of precision, though not all actual water areas are identified as such. Three major aspects are responsible for producing values below 80.0 %: 1) The waterbody shown in the image is not represented by one connected component, but split into parts by artefacts like street lamps, because of the photographer's perspective. Thus, some parts of the waterbody are left out of consideration (Fig. 4g). 2) Images captured between 18.30 h and 19.40 h show shadowing effects on the water's surface due to the zenith angle of the sun, which negatively affected the classification. 3) Shallow water or wet surfaces, such as on the wall of a building or bridge pillars caused by waves, are less likely to be classified correctly (Fig. 4h). Generally, images of higher resolution and taken from a frontal perspective in relation to the object of interest, as well as with higher contrast and brightness, tend to yield a better outcome during segmentation.

The overall segmentation results of the applied image classification workflow showed similar results to those reported by Bruinink et al. (2015:429), with an average precision of 99.2 % and a recall of 91.0 %. They studied nine images captured by experts in order to derive staff gauge measurements from these images. The aim of the approach presented here, however, was to make use of user-generated flood images in order to extract the water level at urban structures and, thus, to handle a much broader range of input images. Crowdsourced image pre-processing, such as pre-selecting those flood images which are most suitable, could be beneficial to the outcome, since it can help to eliminate unsuitable images of low contrast or brightness as well as blurry ones (Lo et al., 2015).





### 5.3 2D-3D mapping

The 2D-3D mapping process results in a reconstructed set of 3D point coordinates indicating the flood level that is represented by the 2D waterline shown in the flood image. The proposed method allows reconstruction of a 3D point for each pixel and, thus, a dense 3D representation of the 2D waterline independent of the given point density of the photogrammetric 3D point
cloud.

The performance of the proposed 2D-3D mapping approach is influenced by the estimation of camera parameters done in the course of the CPR process. A low accuracy for the derived camera location and orientation will consequently result in a low accuracy for the waterline reconstruction. Also, objects in front of the considered façade (e.g., cars or street lamps) can lead to misplacement of 3D water-level points because the 2D waterline along these artefacts does not lie within the same 3D plane
as the relevant façade. They will be erroneously projected onto the façade plane during the 2D-3D mapping process.

Hence, the individual image characteristics are influencing factors insofar as they 1) influence the CRP process and thus the estimation of camera location and orientation, and 2) the clearness of the 2D waterline pixels. Some image characteristics (e.g., artefacts in the foreground of the image) are more troublesome than others when seeking to obtain reliable and accurate results. These could be filtered via user interaction, by making use of participatory sensing or through crowdsourced approaches
(Albuquerque et al., 2016).

### 5.4 Flood elevation determination

The complete set of 3D water-level points sums up to 10,347 individual measurements representing the water surface along the building's façade. These points are based on ground truth data from all input images (n = 66). The median flood elevation $Z_{w,GT}$ is 153.78 m a.s.l. with a mean deviation from the median (MD) of $\pm$ 0.08 m (Fig. 5a). In comparison to the TLS reference
flood elevation value and under consideration of error propagation, the overall accuracy of the derived flood elevation measurements is given as 0.05 m $\pm$ 0.13 m. Figure 5b shows that more than 80 % (54 of 66 images) of the images result in a median water-level elevation within the range of the TLS reference (153.83 m a.s.l. $\pm$ 0.10 m). Only four images result in flood elevation values outside the wave movements of $\pm$ 0.10 m from the derived median flood elevation ($Z_w$). The slightly lower accuracy of 0.05 m can partly be explained by the geo-referencing accuracy of the CRP point cloud, with a median C2C of
0.02 m.

With regard to the natural undulation and waves of the water surface ($\pm$ 0.10 m), the accuracy of $\leq$ 0.10 m obtained from the derived flood elevation values is considered a satisfying result. In comparison, Smith et al. (2014) derived a mean absolute difference of 0.29 m between high-water marks indicated in a photogrammetric point cloud and differential Global Navigation Satellite System (dGNSS) measurements, whereby the water marks were derived from two or more images at a time. The
proposed method in our study, however, only requires one flood image at a time and, thus, allows more flexibility in terms of flood image collection.



## 5.5 Building inundation depth determination

The final inundation-depth results are achieved after calculating the elevation difference between the water-level points and the DTM. The derived depths depend equally on the accuracy of 1) the water-level points and 2) the topography of the respective terrain. Since the topography might change considerably along the façade (e.g., a declining road) but not the water level, descriptive statistics, such as minimum, maximum, or mean inundation need to be taken with caution. The determined depths give rather selective measurements at certain positions along the building's facade.

The inundation-depth findings are compared to the manual field measurements. This results in an accuracy of 0.13 m ± 0.10 m for 533 points lying within the ranges of the seven reference positions. Additionally, expert measurements taken in the TLS point cloud are evaluated in the same way. The overall accuracy of the inundation depth derived by the experts is 0.07 m ± 0.09 m for 56 points in total (Fig. 6). Generally, the expert measurements show slightly more-accurate depth results than the ones derived from the automatic method, especially for positions 1, 6, and 7, where only few terrain points can be found and thus outlier values are more influential. Experts, on the contrary, can account for micro-topography or other irregularities in order to avoid miscalculations due to artefacts or data gaps. Furthermore, it has to be considered that both the in-field measurements as well as the expert measurements are based on the same flood image, whereas the results demonstrated by the automatic approach rely upon a series of 66 different flood images indicating slightly different flood elevations due to waves. All in all, it can be concluded that experts can be better at incorporating irregularities due to their experience. However, computer-based measurements benefit from a more systematic, objective, and reproducible approach that is not subject to human error and interpretation. Furthermore, it is found that the automatic approach can nonetheless achieve a similar accuracy to that of the experts but with the additional advantage of being much more time-efficient. A combination of both approaches, for example a standardized automatic approach with interactive user input for quality assessment, could be a beneficial enhancement of the methodology. In comparison to building inundation depth estimations based on high-resolution SAR data with accuracy values between 0.24–0.81 m (Iervolino et al., 2015), the results given in this study suggest that user-generated flood photographs can serve as an alternative or complementary data source for local building inundation depth determination at a smaller scale.

## 6 Conclusion

We developed a method to derive the local inundation depth within a 3D point cloud based on user-generated flood images. The aim was to determine the accuracy of this proposed workflow concerning the derived flood elevation as well as the resulting inundation depths at a local building scale.

The results of this study have shown that the developed methodology is able to obtain measurements of local flood elevation and building inundation depth to within an accuracy of < 0.20 m. The overall accuracy is 0.05 m ± 0.13 m for flood elevation and 0.13 m ± 0.10 m for the local building inundation depth. It is also shown that the method is applicable for crowd-sourced images captured in an unorganized manner. Moreover, the measurements taken by experts revealed that the proposed method



produces results almost as accurate as those provided by human experts. The main advantage of the semi-automatic segmentation process is its time efficiency and, thus, the possibility of processing multiple flood images to receive more robust inundation results.

The key findings of this study can be summarized in the following points. 1) A satisfactory accuracy of local flood elevation and building inundation depth determination can be achieved using the proposed workflow. Under consideration of the natural fluctuation of the water surface (here $\pm$ 0.10 m), the final overall precision of the method ($\pm$ 0.13 m) is only slightly less precise than the inherent uncertainty of the phenomenon itself. 2) The extraction of the 2D waterline from the provided flood image has a major influence on the accuracy of the final results. It is, thus, recommended that the image segmentation process be stabilized by pre-selecting the available flood images according to their individual image characteristics. Images with low contrast, especially in wet areas along the façade, tended to result in a less accurate 2D waterline.

In comparison to other studies using, for example, high resolution SAR data for inundation depth determination, it has been shown that user-generated images can serve as an alternative or complementary data source to examine the effects of flooding on a very local scale. Our approach is, thus, considered beneficial for applications such as flood damage assessment, or resilience planning, and more generally all research dealing with urban floods. Moreover, our study delivers a low-cost approach for automatically detecting the flood elevation and inundation depth indicated in a flood image in 3D, and does not rely on in situ estimations. This allows the analysis of a much broader data set while not necessarily requiring field work. Furthermore, the methodology can be adapted to various other use cases where only singular image information is given.

*Author contributions*. All authors worked on the idea and the framework of this study. L. Griesbaum and S. Marx did data capturing in the field. L. Griesbaum carried out the experiments and developed the code model. L. Griesbaum prepared the manuscript with contributions of B. Höfle. All authors read and approved the final paper.

*Competing interests*. The authors declare that they have no conflict of interest.

*Acknowledgements*. This work was supported by the Heidelberg Institute for Geoinformation Technology (HeiGIT) and the Heidelberg Academy of Sciences and Humanities (HAW).

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





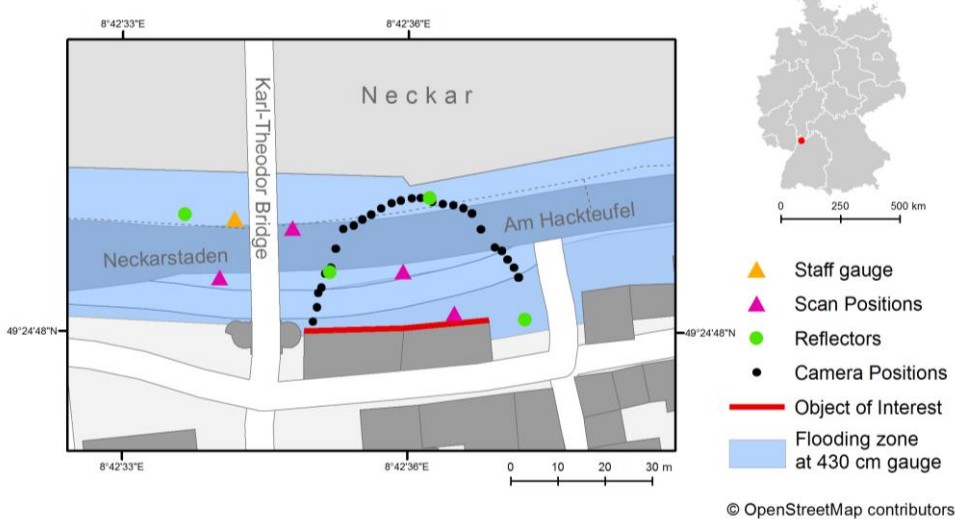

**Figure 1: Overview map of the study area, including the measurement set-up for acquisition of the terrestrial laser scanning (TLS) data and the camera positions of the non-flood images.**





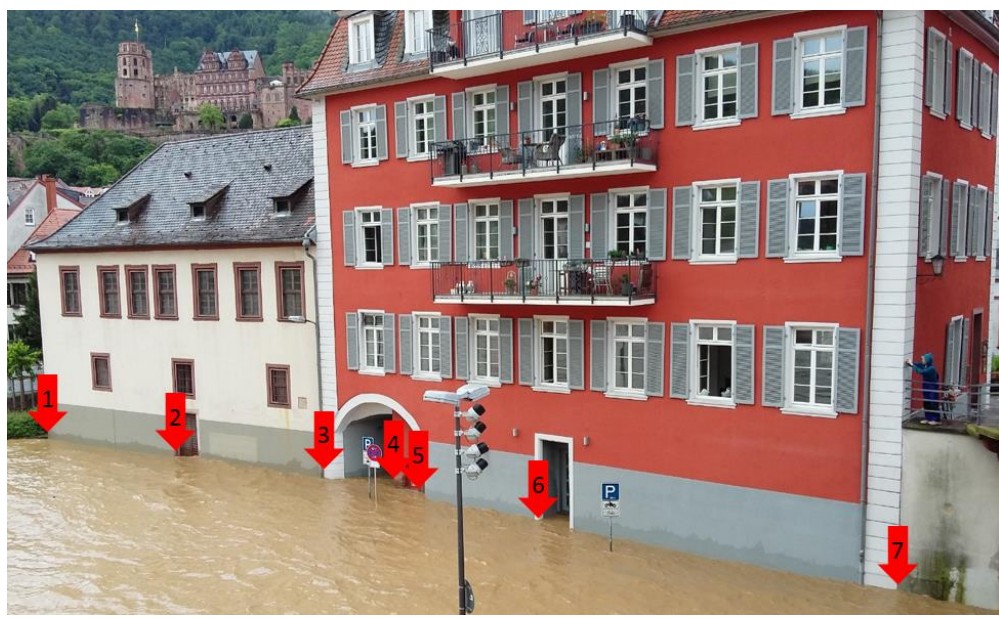

**Figure 2: Example flood image with reference positions for manual in-field inundation depth measurements at the study site indicated with red arrows.**





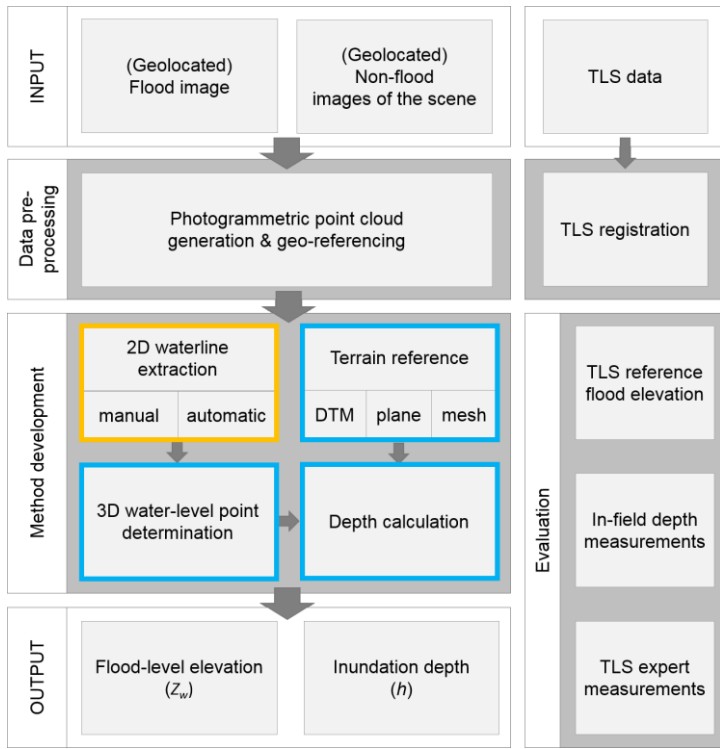

**Figure 3: Workflow of this study, depicting the individual operations of the proposed methodology. Blue indicates operations executed on point cloud level (3D), yellow indicates operations executed on image level (2D).**





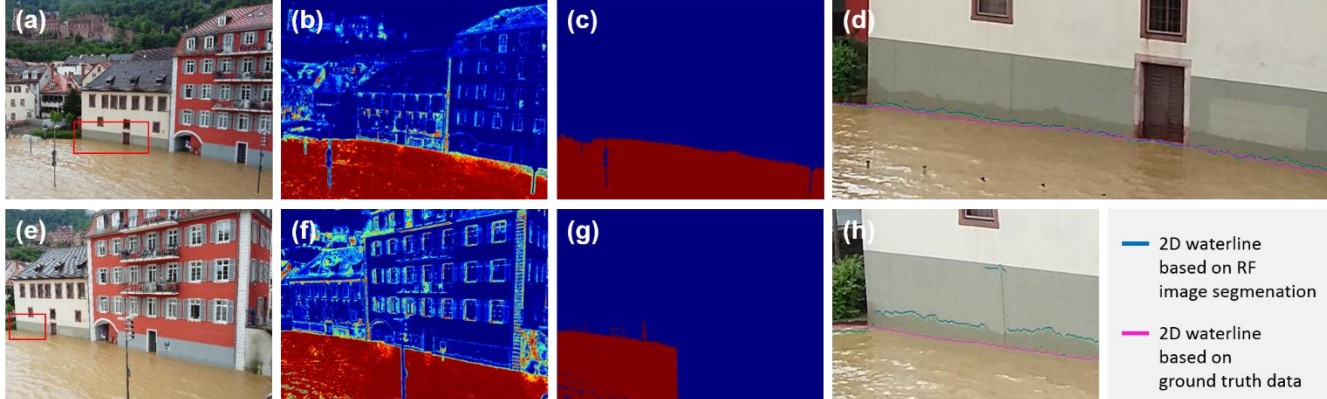

**Figure 4: Segmentation results of two different flood images (top line: precision = 99.6 %, recall = 95.6 %, bottom line: precision = 98.5 %, recall = 54.3 %). (a and e) depict the original images; (b and f) show the probability maps after RF classification; (c and g) are the binary images of the finally classified waterbody after largest component analysis. (d and h) show subsets of the extracted 2D waterlines.**





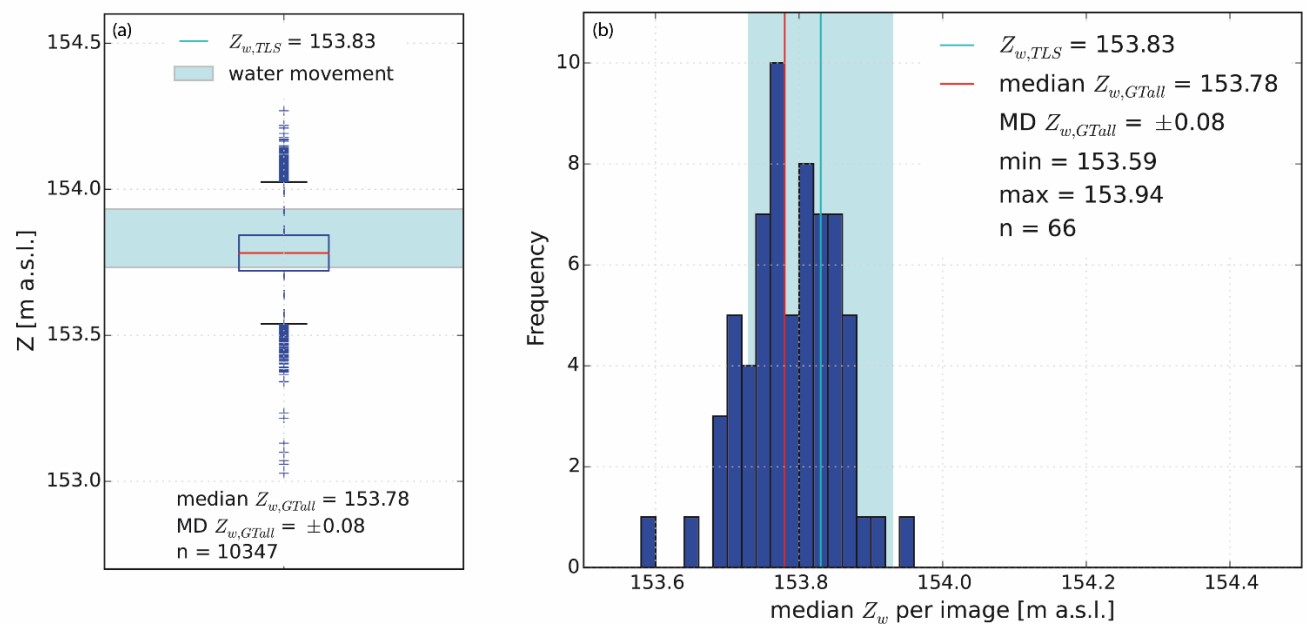

**Figure 5: Distribution of flood elevation values in comparison to the TLS reference flood elevation value $Z_{w,TLS}$ from all input flood images based on (a) all single points and (b) aggregated per image.**





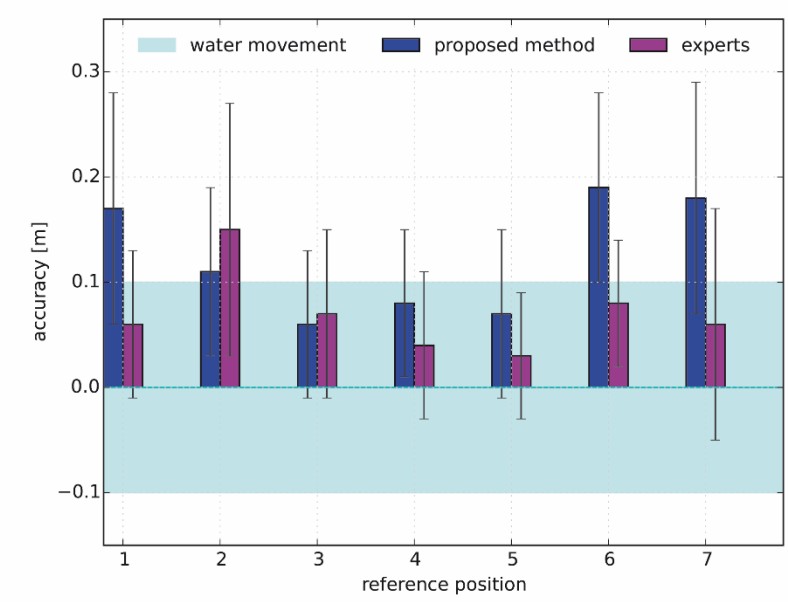

**Figure 6: Derived inundation depth accuracy and precision of the proposed method (blue) and expert measurements (magenta) given in comparison to the water movement (cyan).**





**Table 1: Manual in-field inundation depth measurements at the seven reference positions at the study site, as given in Fig. 2.**

| Reference position | Manual in-field inundation depth measurement [m] |
|---|---|
| R1 | 0.55 |
| R2 | 0.70 |
| R3 | 0.94 |
| R4 | 0.96 |
| R5 | 1.02 |
| R6 | 1.17 |
| R7 | 1.17 |





**Table 2: Quality indicators of the photogrammetric point cloud registration in comparison to the TLS point cloud.**

| Quality indicator | TLS | | Photogrammetric approach | |
|---|---|---|---|---|
| Median C2C distance [m] | Reference | | 0.02 | |
| | **Facade** | **Terrain** | **Facade** | **Terrain** |
| **Completeness [%]** | 93.2 | 66.8 | 87.6 | 37.6 |
| **Mean point density [points per 0.2 m x 0.2 m raster cell]** | 1,909.4 | 822.8 | 91.2 | 18.3 |