# Peer review of "Direct local building inundation depth determination in 3D point clouds generated from user-generated flood images"

_Natural Hazards and Earth System Sciences, 2017_

## Referee Comment (RC1) · Anonymous Referee #1 · 19 Apr 2017

The paper is very interesting and well written. It effectively concludes that the methods are technically sound and deliver outcomes within the normal limits of photogrammetry and the flood hazard.

However the method is relatively complex to implement in practice and its value beyond conventional inundation mapping techniques is not well positioned. In practice, the 2D Floodline is the major information require by hydrologists, engineers and insurance assessors, so the value in obtaining further information in the 3D context needs to be elaborated. It would also benefit with some further discussion on potential automation approaches and perhaps the use of google street view or other data sets to assist in the automation.

[Figure]

Overall it is a solid paper but some stronger justification for the need for this work would make it a better paper.

---

## Referee Comment (RC2) · G. J.-P. Schumann (Referee) · 30 Apr 2017

This paper describes the use of smart-phone (or hand-held) camera-acquired flood and non-flood images to extract waterlines from buildings and extract flood water height as well as inundation depths.

It was a real pleasure to read this paper. It is technically innovative, very well written and easy enough to follow even for a non-technical reader, and it gives a lot of references and a complete literature review at the beginning which I appreciate very much. The figures are of excellent quality.

Actually, it is sad to see that such papers (referring to its format and structure as well

as style) are becoming a rare bread among the overwhelming literature nowadays. I think this paper should be revised with attention to only a few minor comments:

- In the abstract, the authors should refer to the actual accuracy of the results obtained

- In the introduction, the authors mention the application of flood disaster response and recovery. In my opinion, the authors should emphasize this goal/application more and I also think therefore the authors should try later on in the paper to give to refer to timeliness of the processing. How long does their method take to generate the information?

- Also, is there an open-source style software bundle that processes images into the information for response, recovery teams etc? I completely understand if the later is not available but are there any plans to do that? Given that the whole processing chain is quite technical but it is aimed at non-technical applications too, I think this is an important step to consider

---

## Author Comment (AC1) · 9 May 2017

Dear colleague,

Thank you for your valuable comments on our manuscript.

Our research is meant to show the applicability and benefits of user-generated flood images for the purpose of documenting local building inundation in cases where either traditional information gathering systems are inapplicable, or we lack adequate reconstruction tools to process the data that exists. Urban floods are especially difficult to document. A 2D floodline often does not suffice to depict the extent of flooding in an urban area because of the highly complex structure and topology of cities and towns.

The need for both spatially and temporally high-resolution data presents a particular challenge in connection with measurement of urban floods. Due to the increasing distribution of information technologies and the correspondingly increasing virtual participation in flood mitigation activities, many – still difficult-to-document – urban flood events are now being indirectly documented by means of user-generated, geotagged flood images (Fazeli et al., 2015). So far, common flood reconstruction methods have lacked the ability to extract local flood information from such images. Our method, however, does enable the extraction of relevant inundation depth values for flooded buildings depicted in these images. In this sense, we understand this publication as proof of concept, which can yet be further automated in order to make it an operational tool for non-technical users.

The major benefit of this 3D approach in comparison to 2D approaches for flood level evaluation lies in its ability to determine inundation depths at single objects/buildings. This information is valuable for building damage assessment. Many flood loss models are based upon – amongst other parameters – a depth-damage function. For example, one typical model for damage assessment is the so-called Flood Loss Estimation Model for the Private Sector (FLEMOps), which is employed for micro-scale applications (i.e. at the building level). Besides building type and quality, one major input parameter for that model is the inundation depth of single buildings (Thieken et al., 2008). Moreover, this third dimension of flood information could be beneficial to local authorities by complementing manual measurements, which are usually captured only on the basis of visual flood markers, such as mud lines at building façades, to provide records on the impact of the event. This newly acquired inundation data of single buildings could then be used for further flood simulation, flood-map generation, or flood risk analysis.

We agree that an implementation of the proposed workflow in form of a (web) service or mobile app would be highly beneficial for local authorities, disaster managers, engineers, and insurance assessors. Eye witnesses may upload flood images taken during

or in the immediate aftermath of a flooding event, which could yield a large data set of inundation depth values. Furthermore, the integration of already existent data sources, such as Mapillary (https://www.mapillary.com/), which provides large sets of non-flood images, is feasible, however requires further research.

Best regards,

Luisa Griesbaum, Sabrina Marx and Bernhard Höfle

References

Fazeli, H. R., Nor Said, M., Amerudin, S., and Abd Rahman, M. Z.: A Study of Volunteered Geographic Information (VGI) Assessment Methods for Flood Hazard Mapping: A Review, Jurnal Teknologi, 75, 127–134, doi:10.11113/jt.v75.5281, 2015.

Thieken, A. H., Olschewski, A., Kreibich, H., Kobsch, S., and Merz, B.: Development and evaluation of FLEMOps – a new Flood Loss Estimation MOdel for the private sector, in: International Conference on Flood Recovery, Innovation and Response, London, England, 02.-03. July 2008, 315–324, 2008.

---

## Author Comment (AC2) · 16 May 2017

Dear Dr. Schumann

Thank you for your valuable comments. We very much appreciate your feedback.

The results of our methods are evaluated for their accuracy with respect to both the derived TLS measurements for the flood elevation and manual in-field measurements of building inundation depth. The accuracy is calculated as the difference to the reference measurement and is always stated together with the corresponding measurement's uncertainty. Accordingly, as we mention in the abstract, the accuracy obtained for the derived flood elevation is 0.05 m $\pm$ 0.13 m, while the accuracy of the derived

inundation depth is 0.13 m ± 0.10 m.

The proposed workflow is meant to support flood management by providing data on flood parameters, such as flood elevation and building inundation depth, in cases where traditional data sources may not provide sufficient information. Such information is needed for disaster managers to assess and minimize the impact of disastrous events, for example to facilitate damage assessment or risk analysis. Our method can be applied immediately after a flood event occurs, as soon as a flood image and several non-flood images become accessible. In the case of disaster response, non-flood images will ideally be readily available, whether due to preventive measurement taking or from other data sources. Otherwise, one must wait until the water has subsided in order to capture the required non-flood images. Because no specific qualifications are necessary for image acquisition, anyone can contribute to this form of flood documentation. As a benefit of close-range sensing, the photographer can capture both flood and non-flood images at a safe distance from the inundated object.

The calculation time of our method depends on the amount of images to process and the available computational resources. Once the photogrammetric point cloud has been calculated, in our case with an ordinary machine (3.60 GHz CPU, 8 cores, 32 GB RAM), the calculations take about five minutes for one building. The good news is that our approach can be scaled easily by improving computing power on a single machine (e.g. number of cores) or by using distributed parallel computing (e.g. through cluster computing or cloud services) because single buildings can be processed independently of one another. It is important to bear in mind that this is our personal experience: Computing performance has not been assessed under controlled conditions. We do however assume that near real-time applications could be feasible if base datasets already exist.

So far, the code has not been made publically available because – after scientific validation and publication in NHESS – we aim to improve the code from a technical standpoint (e.g. improve performance and modularity). However, the implementation of the
method in the form of a (web) service or mobile app is being considered and, thus, of interest to further investigations. Non-technical users, especially, stand to benefit from such a service, since it could easily provide flood measurements based on just the required flood and non-flood images. For anyone interested, we are happy to collaborate and look forward to seeing this technology used in a multitude of applications in the future.

Kind regards,

Luisa Griesbaum, Sabrina Marx, Bernhard Höfle

---

## Author Response (AR1)

Dear Prof. Dr. Bruno Merz,

Thank you very much for your decision on publishing our manuscript "Direct local building inundation depth determination in 3D point clouds generated from user-generated flood images" with minor revisions.

We revised the manuscript according to the referee comments given during the discussion phase. We hope that by including the remarks, the readability of the paper increases, and the focus of the study becomes clearer. We think that emphasizing the goal in disaster response and recovery in the introduction as well as in the conclusion parts makes reading the paper more interesting and valuable.

Below we elaborate point-by-point on the changes made in the manuscript.

We are looking forward to your opinion on the revised version and your decision about publishing the manuscript in NHESS.

Kind regards, Luisa Griesbaum, Sabrina Marx, Bernhard Höfle

**Referee Comment #1:**

"However the method is relatively complex to implement in practice and its value beyond conventional inundation mapping techniques is not well positioned."

This study is meant as proof of concept to show a feasible method for the derivation of flood information from non-structured user-generated flood images (p.11, l.4f). We enhanced both the introduction (p.3, l.23ff) and the conclusion (p.11, l.27f) in order to stress the value of our method as complementary and supportive technique for disaster managers.

"In practice, the 2D Floodline is the major information require by hydrologists, engineers and insurance assessors, so the value in obtaining further information in the 3D context needs to be elaborated."

In order to emphasis the benefits of 3D data for flood management in the case of urban flooding, namely the inundation depth and flood elevation, we added some additional explanation on this in the introduction part (p.2, l.25ff).

"It would also benefit with some further discussion on potential automation approaches and perhaps the use of google street view or other data sets to assist in the automation."

Indeed, implementation and thus further automation of the developed workflow in form of a (web) service could be highly beneficial for disaster managers. This, as well as the integration of other data sources, should be subject of further investigations (p.11, l.27ff).

**Referee Comment #2:**

"In the abstract, the authors should refer to the actual accuracy of the results obtained"

The developed method is evaluated on basis of 66 iterations with different flood images showing the same building. The accuracy of both the water level, and the inundation depth is calculated as the difference between the derived results and the reference data. Both accuracies are given with the corresponding measurements' uncertainty. These values,  $0.05 \text{ m} \pm 0.13 \text{ m}$  for the flood elevation and  $0.13 \text{ m} \pm 0.10 \text{ m}$  for the derived inundation depth, are stated in the abstract (p.1, 1.15f).

"In the introduction, the authors mention the application of flood disaster response and recovery. In my opinion, the authors should emphasize this goal/application more and I also think therefore the authors should try later on in the paper to give to refer to timeliness of the processing. How long does their method take to generate the information?"

We added some additional information on how our approach can effectively support the work of local authorities or disaster managers in order to stress the value of our approach for flood disaster response and recovery (p.3, l.23ff).

Moreover, in the discussion part we do now refer to the timeliness of the processing and stress the possibility of almost real-time application (p.11, l.31f).

"Also, is there an open-source style software bundle that processes images into the information for response, recovery teams etc? I completely understand if the later is not available but are there any plans to do that? Given that the whole processing chain is quite technical but it is aimed at non-technical applications too, I think this is an important step to consider"

Our complete workflow can be stepwise reproduced and is implemented with open source software components (e.g. VisualSFM, Python scripting). Our aim for the future is indeed to put these components together in an easy-to-use web service. At the moment we do not have an operational software available because in our pioneer study we focused on assessing the general feasibility rather than developing the software.

[revised manuscript text omitted]